# The Metabolic Changes of Artesunate and Ursolic Acid on Syrian Golden Hamsters Fed with the High-Fat Diet

**DOI:** 10.3390/molecules25061392

**Published:** 2020-03-18

**Authors:** Shichen Pu, Yumin Liu, Shan Liang, Pin Liu, Hongmei Qian, Qian Wu, Yuliang Wang

**Affiliations:** 1Plant Biotechnology Research Center, Fudan-SJTUNottingham Plant Biotechnology R&D Center, School of Agriculture and Biology, Shanghai Jiao Tong University, Shanghai 200240, China; pushichen@sjtu.edu.cn (S.P.); auduna@sjtu.edu.cn (S.L.); liupinshe@163.com (P.L.); hmqian@sjtu.edu.cn (H.Q.); 2Instrumental Analysis Center, Shanghai Jiao Tong University, Shanghai 200240, China; ymliu@sjtu.edu.cn; 3Shanghai Center for Bioinformation Technology, Shanghai Industrial Technology Institute, Shanghai 201203, China; wuqian@scbit.org

**Keywords:** Artesunate, ursolic acid, hyperlipidemia, metabolomics

## Abstract

Artesunate was well known as an antimalarial drug. Our previous research found that it has hypolipidemia effects in rabbits fed with a high-fat diet, especially combined with ursolic acid. In this study, we reconfirmed the lipid-lowering effect of artesunate and ursolic acid in hamsters and analyzed the metabolic changes using gas chromatography time-of-flight mass spectrometry (GC/TOF MS). Compared with the model group, a variety of different metabolites of artesunate and ursolic acid, alone or in combination, were found and confirmed. These differential metabolites, including fatty acids, lipids, and amino acids, were involved in lipid metabolism, energy metabolism, and amino acid metabolism. It indicated that two agents of artesunate and ursolic acid could attenuate or normalize the metabolic transformation on these metabolic pathways.

## 1. Introduction

Modern research shows that hyperlipidemia promotes the formation of nonalcoholic fatty liver, atherosclerosis, coronary heart disease, and is often accompanied with diabetes [1,2,3,4]. With the transformation of diet structure, the incidence of hyperlipidemia has increased worldwide.

Many phytochemicals—such as phytosterol, berberine, and curcumin—can contribute to ameliorating hyperlipidemia-related diseases [5,6,7]. Artesunate is a semisynthetic derivative of artemisinin and has efficient anti-malaria effects [8,9,10]. Recent studies have revealed that artesunate (ART) has clear anti-tumor activity, suggesting that it could be a good candidate as a chemotherapeutic agent [11,12]. Ursolic acid (UA) is a pentacyclic triterpene, which is widely distributed throughout the whole plant kingdom, and is a major component of some traditional medicinal plants [13]. It has been found to have anti-cancer and anti-hyperlipidemia effects [14,15,16]. In our previous study, we reported the synergistic effects of artesunate and ursolic acid on lipid reduction and anti-atherosclerosis [17]. Their synergy mechanism is unclear.

With the increasing attention of lipid research, lipidomics, a sub-discipline of metabolomics, has become a hot spot in scientific research. Lipidomics is the large-scale profiling and quantification of biogenic lipid molecules and involves comprehensively studying lipid pathways and interpreting their physiological significance based on analytical chemistry and statistical analysis [18,19]. Lipidomics will not only provide insight into the physiological functions of lipid molecules, but will also provide an approach to discovering important biomarkers for diagnosis or treatment of human diseases [20]. The application of lipidomics supplies new methods for investigating the action mechanism of lipid-lowering drugs. The research confirmed that gas chromatography/mass spectrometry (GC/MS)-based metabonomic profiling is effective for studying the lipid-regulating effects of Ginkgo biloba extract on diet-induced hyperlipidemia in rats [21].

In this study, we once again confirmed the lipid-lowering effect of artesunate and ursolic acid on a different animal model. Moreover, using a gas chromatography time-of-flight mass spectrometry (GC/TOF MS) method, we explored the changes in lipid metabolism caused by artesunate and ursolic acid. The finding will contribute to understanding the synergism mechanism of the lipid-lowering effects treated by ursolic acid and artesunate.

## 2. Results

### 2.1. Artesunate/Ursolic Acid Combination Therapy Has Better Lipid-Lowering Effect than Atorvastatin

As shown in Figure 1, TC, TG, LDL-c in the serum of the model groups that were given high-fat diets were significantly higher than the control groups, which were provided with standard chow, suggesting that the high-fat model was successfully established. Compared with the model group, atorvastatin, artesunate, and combination significantly reduced TC, TG, and LDL in the serum of golden hamsters (Figure 1a–c), and artesunate and ursolic acid monotherapy, or combination of both, can increase the ratio of high-density lipoprotein to low-density lipoprotein (Figure 1d). The high-dose combination, groups 2 and 3, were the only two that could increase the apoA-I content of serum (Figure 1e). 

### 2.2. Artesunate/Ursolic Acid Combination Therapy Has Lower Liver Toxicity Compared with Atorvastatin

The acquired results of AST and ALT showed that atorvastatin caused great damage to the liver function of hamsters (Figure 2). Artesunate/ursolic acid monotherapy did not significantly increase the concentrations of AST and ALT, their combination can even reduce serum concentrations of AST and ALT compared with those in the model group. 

### 2.3. Metabolic Difference between Drug Treatment and Model Group

The serum samples of four groups (artesunate group (20 mg/kg), ursolic acid group (20 mg/kg), combination group (artesunate 20 mg/kg + ursolic acid 20 mg/kg), and model group) were analyzed by GC/TOF MS. The OPLS-DA analysis was performed to get a deeper look into the differential metabolites, which are accountable for the intergroup separation. The parameters for evaluating the OPLS-DA model are R^2^Y and Q^2^, which represent the interpretation rate and prediction rate of the model, respectively. As shown in Appendix A (see Appendix A), the modeling parameters indicated that the model establishment was effective. It could be clearly observed that metabolites in each group can be better clustered together, and the differences within the group are significantly reduced (Figure 3). 

In order to prevent the model from overfitting, the quality of the model was examined by seven-cycle interactive verification and 200-response ranking test. The validation plots of the OPLS-DA models were listed in Figure 4, generated from the tests that were randomly permuted 900 times with the corresponding predictive principal component. The green square was R2 (cum), denoting the explained variance of the model. The blue diamond was Q2 (cum), standing for the predictive ability of the model. These results assessed that these OPLS-DA were not overfitting and indicated that these OPLS-DA had high separating capacity. 

The differential metabolites were shown in Figure 5. As shown in the figure, metabolites are sorted from top to bottom based on the value of Fold change (Fc). Fc is the ratio of the mean value for measured samples obtained from each drug treatment group to the mean value for the measured samples obtained from the model group. Fc (> 1.2) indicates a relatively higher concentration present in each drug treatment group as compared to the model group, whereas a value of < 0.8 means a relatively lower concentration. Compared with the model group, there were 19 differential metabolites in the artesunate group, eight in the ursolic acid group, and 28 in the combination group. It indicated that the combination group affected more metabolic pathways. 

As shown in Figure 6, cholesterol and triglycerides were significantly decreased in the three drug-administered groups, which was consistent with the results of the serum biochemical tests (Figure 6). 

Many long chain fatty acids, such as hexadecanoic acid, palmitelaidic acid, arachidonic acid, tetracosanoic, undecanoic acid, and (*E*)-11-Eicosenoic acid were decreased in the artesunate group and the combination group (Figure 7). 

However, only in the combination group, oleic acid, α-linolenic acid, and linoleic acid were decreased compared with the model group (Figure 8). 

Besides lipids, differential metabolites also include amino acids, such as leucine, lysine, and threonine. 

## 3. Discussion

Golden hamsters had some unique advantages as animal models of lipid disorders. First, In terms of cholesterol metabolism, endogenous cholesterol was about 85% originated from the outside of the liver, while it was 90% in humans, and only 35% in male rats [22]. The proportion of cholesterol converted to cholic acid, the mechanism of bile acid metabolism and diet-induced atherogenic lesions in golden hamsters were similar to those in humans [23]. In addition, the mechanism of lipoprotein metabolism, such as the composition of lipids and proteins, was also similar to that of humans. Therefore, golden hamsters were suitable for the hyperlipidemia model [24]. Our current studies reveal a significant lipid-lowering effect of artesunate and ursolic acid in golden hamsters. There was no significant difference in the serum ALT and AST levels, which proved that artesunate and ursolic acid had no damage to liver function. In previous studies, ursolic acid showed a significant reducing effect on cholesterol and triglyceride in the serum of rats, which agreed with our current studies [25].

Lipomics analysis revealed the change of many fatty acids in the serum of the drug treatment groups. The down-regulated fatty acids were long chain fatty acids, such as undecanoic acid, tetracosanoic acid, linoleic acid, alpha-linolenic acid, arachidonic acid, oleic acid, and palmitic acid. However, short chain fatty acids, such as acetic acid and butyric acid were upregulated in the plasma, compared with the model group. We speculated that artesunate and ursolic acid could promote the decomposition of long chain fatty acids into short chain small molecule fatty acids. Cholesterol is metabolized by the enzyme CYP7A1 to produce cholic acid [26]. The content of cholic acid was upregulated compared with the model group, which indicated that artesunate and ursolic acid could promote the conversion of cholesterol to cholic acid.

Pyruvate was a key metabolic intermediate of glucose metabolism (glycolysis and gluconeogenesis) and the tricarboxylic acid cycle (TCA). Pyruvate could be converted into acetyl-coenzyme A through the reaction of decarboxylation. Cortez et al. found that the use of diets supplemented with pyruvate and dihydroxyacetone (DHA-P) significantly reduced triglyceride levels in the serum of obese rats [27]. In the absence of oxygen, pyruvate was converted to lactic acid under the catalysis of lactate dehydrogenase (LDH). Moreover, the increase of the level of lactic acid could aggravate the beta-oxidation of fatty acids in the body. After the intervention of artesunate, the level of lactate and pyruvate in the serum both increased significantly, indicating that artesunate could affect the energy metabolism.

In addition to fatty acids, amino acids are also an important class of differential metabolites. Artesunate can obviously increase the level of glutamate, threonine, lysine, and leucine.

Glutamate were precursors of the natural antioxidant glutathione (GSH). Glutathione (GSH) is an important antioxidant in cells, whose main function is to protect biofilms, nucleotides, and various proteins from free radical damage. Therefore, it can be speculated that artesunate may have a protective effect on lipid peroxidation and oxidative stress induced by a high-fat diet.

Under the catalysis of threonine dehydrogenase (TDH), threonine was oxidized to 2-amino-3-oxybutyric acid and rapidly metabolized to glycine and acetyl-coA [28], while lysine was a precursor substance of carnitine. The acetyl-coA and carnitine can promote the β-oxidation of fatty acids in the mitochondria. Coincidentally, Mong et al. pointed that the addition of lysine in mouse feed can alleviate the formation of hyperlipidemia [29]. Sun et al. reported that leucine could block the activity of proteins related to the transport and synthesis of fatty acids; thereby, inhibiting the synthesis of fatty acids [30]. As a result, it was presumed that artesunate could promote the β-oxidation of fatty acids and inhibit the synthesis of fatty acids.

## 4. Materials and Methods

### 4.1. Chemical and Reagent

Forty-eight Syrian golden hamsters (male) were obtained from the Songlian Experimental Animal Farm (Songjiang District, Shanghai). The animal certificate number is SCXK (Shanghai, China) 2012-0011. Artesunate (batch number 000129) was purchased from Guangzhou Hanfang Pharmaceutical Co. (Guangzhou, China). Ursolic acid (batch number XC091225) was purchased from Xi’an Xiaocao Plant Technology Co. (Xi’an, China). Atorvastatin calcium (batch number: 124F027116) was purchased from Wuxi Shengfu Pharmaceutical Co. (Wuxi, China).

### 4.2. The Feeding Conditions and Treatments for Syrian Golden Hamsters

The golden hamsters were housed in a positive pressure purified and ventilated animal room. Room temperature and humidity of the animal room were at 21 ± 1 °C and 60 ± 10%, respectively, using artificial lighting to simulate the alternating day and night. In addition, the golden hamsters were provided with standard golden hamster chow and tap water, ad libitum. Forty-eight male golden hamsters were randomly divided into 8 groups, 6 in each group: normal group; model group; positive control group (atorvastatin group, 10 mg/kg); artesunate group (20 mg/kg); ursolic acid group (20 mg/kg); combination group 1 (artesunate 10 mg/kg + ursolic acid 10 mg/kg); combination group 2 (artesunate 20 mg/kg + ursolic acid 20 mg/kg); combination group 3 (Artesunate 40 mg/kg + ursolic acid 40 mg/kg). Except for the normal group, the other groups were given high-fat diets to establish a plasma high-fat model of golden hamsters, and received corresponding treatment agent via gavage administration at the designed dose. The composition of the high-fat diet used in this experiment is listed as follows: lard 10%, cholesterol 2%, propylthiouracil 0.2%, pig bile salt 0.5%, and basic feed 87.3%. Serum samples were collected after two months of feeding, taken from the same part of the eyelids. Afterwards, the serum samples were separated by centrifugation. The lipid related indicators, such as total cholesterol (TC), triglyceride (TG), high-density lipoprotein (HDL-c), low-density lipoprotein (LDL-c), apolipoprotein A-I (apo A-I), and the liver function related indicators, such as alanine aminotransferase (ALT) and aspartate aminotransferase (AST), were subsequently measured. Data from the serum biochemistry determinations were expressed in the form of mean ± SD. Statistical analysis was conducted using the two-tailed Student’s t-test. A *p*-value of less than 0.05 was considered statistically significant.

### 4.3. Preparation of Serum Samples

All of the animal procedures and experiments were carried out according to the National Institutes of Health Guide for the Care and Use of Laboratory Animals, and had been approved by the Bioethics Committee of Shanghai Jiao Tong University. The serum samples were taken using a capillary fundus venous plexus. The serum samples were allowed to stand at room temperature for 30 min. After the serum samples were coagulated, they were centrifuged at 4 °C for 15 min, and the rotation speed was 3000 rpm. After centrifugation, the supernatant was transferred to a clean Eppendorf tube and stored in a −80 °C refrigerator, pending for biochemical analysis. Subsequently, the serum samples were sent to Shanghai Aidekang Medical Testing Center for serum biochemical analysis.

### 4.4. Preparation of Samples and Analysis by GC/TOF MS

A 100-μL aliquot of serum sample was added with two internal standards, L-2-chlorophenylalanine (10 μL, 0.3 mg/mL in water) and heptadecanoic acid (10 μL, 1.0 mg/mL in methanol) and vortexed for about 10 s. The mixed solution was extracted with 300 μL of methanol/chloroform (3:1) and vortexed for 30 s, and kept at the temperature of −20 °C for 8 min. Subsequently, the treated samples were centrifuged at 10,000 rpm for 8 min. Next, 300 μL of supernatant was transferred to a glass sampling vial and dried completely at room temperature. Then, we used a two-step method in order to derivatize the residue. Firstly, 80 μL of methoxyamine (15 mg/mL in pyridine) was added to the residue in the glass sampling vial for methoxymation. After standing at 30 °C for 90 min, the samples were spiked with 80 μL of BSTFA (1% TMCS) and the process of trimethylsilylation was performed at 70 °C, lasting for one hour.

Each 1-μL aliquot of the derivatized specimen was injected in the split-less mode into an Agilent 6890 N gas chromatograph (StJoseph, MI, USA), combined with a time-of-flight mass spectrometer (Pegasus HT, Leco Co., StJoseph, MI, USA). Separation was achieved on a DB-5MS capillary column (30 m × 250 μm I.D., 0.25 μm film thickness; (5%-phenyl)-methyl-polysiloxane bonded and cross-linked; Agilent J&W Scientifi, Folsom, CA, USA) with helium as the carrier gas, at a constant flow rate of 1.0 mL/min [31]. To minimize systematic analytical deviations, each analytical batch may contain one spiked sample as quality control and one blank sample run together every 10 injections. The temperature of injection was set to 270 °C. The GC initial oven temperature was set at 80 °C for 2 min. It was raised linearly to 180 °C at a rate of 10 °C/min, and to 240 °C at a rate of 5 °C/min, and to 290 °C at a rate of 25 °C/min afterwards. Finally, the temperature remained at 290 °C for 9 min. The temperature of transfer line and ion source were set at 260 °C and 200 °C, respectively. Electron impact ionization (70 eV) at full scan mode (*m/z* 30–600) was used to obtain mass spectra at a rate of 20 spectrum/second in the GC/TOF MS setting.

### 4.5. Identification of Metabolites

The acquired files from GC/TOF MS analysis were exported to NetCDF format by ChromaTOF software (v3.30, Leco Co., St.Joseph, MI, USA). Metabolite identification was executed by ChromaTOF software combined with the National Institute of Standards and Technology (NIS) mass spectral libraries (NIST 14), and LECO/Fiehn Metabolite mass spectral library (Version 1.00, the lab of Oliver Fiehn at UCD, CA, USA), with a similarity of more than 70%. A majority of the metabolites detected were identified using the reference compounds available.

### 4.6. Data Processing and Statistical Analysis

The computable document format (CDF) files were extracted using custom scripts (revised MATLAB toolbox HAD) in the MATLAB 7.1 (The MathWorks, Inc., Natick, MA, USA) for data pretreatment procedures, such as baseline correction, denoising, smoothing, and alignment; time-window splitting; and peak feature extraction (based on multivariate curve resolution algorithm) [32,33,34]. The resulting three-dimensional data set, which includes sample name, peak retention time, and peak intensity, was normalized for multivariate statistical analysis, according to the published methods [35]. Then, the normalized data matrix was imported into the SIMCA-P + 14.0 software package (Umetrics, Umeå, Sweden), the supervised (orthogonal) partial least squares method was then used to analyze orthogonal partial least squares discriminant analysis (OPLS-DA) to distinguish the overall differences in metabolic profiles between groups, and to find the differential metabolites between groups. In order to prevent the model from overfitting, the quality of the model was censored by seven-cycle interactive verification and 200-response ranking test.

In the OPLS-DA analysis, variable importance in the projection (VIP) greater than 1 is considered to be a difference. The univariate statistical analysis and Student’s t-test experiments were further performed on the metabolites identified from GC/TOF MS. Metabolites with significant differences could be selected, according to the criteria of variable importance in the projection (VIP > 1.0) of the OPLS-DA models and the *p*-values (*p* < 0.05) in the t-tests.

## 5. Conclusions

In this paper, we confirmed that artesunate and ursolic acid have significant lipid-lowering effects on hyperlipidemia golden hamsters. GC/TOF MS combined with multivariate statistics analysis has been used to further explore the mechanism of lipid lowering. Many differential metabolites, including fatty acids and amino acids, were found and identified. The metabolic shifts could help explain that artesunate and ursolic acid effectively achieve the effect of lipid-lowering by affecting some metabolic pathways, such as lipid metabolism, energy metabolism, and amino acid metabolism. Two agents of artesunate and ursolic acid could attenuate or normalize the metabolic transformation with different degrees on these metabolic pathways. 

## Figures and Tables

**Figure 1 molecules-25-01392-f001:**
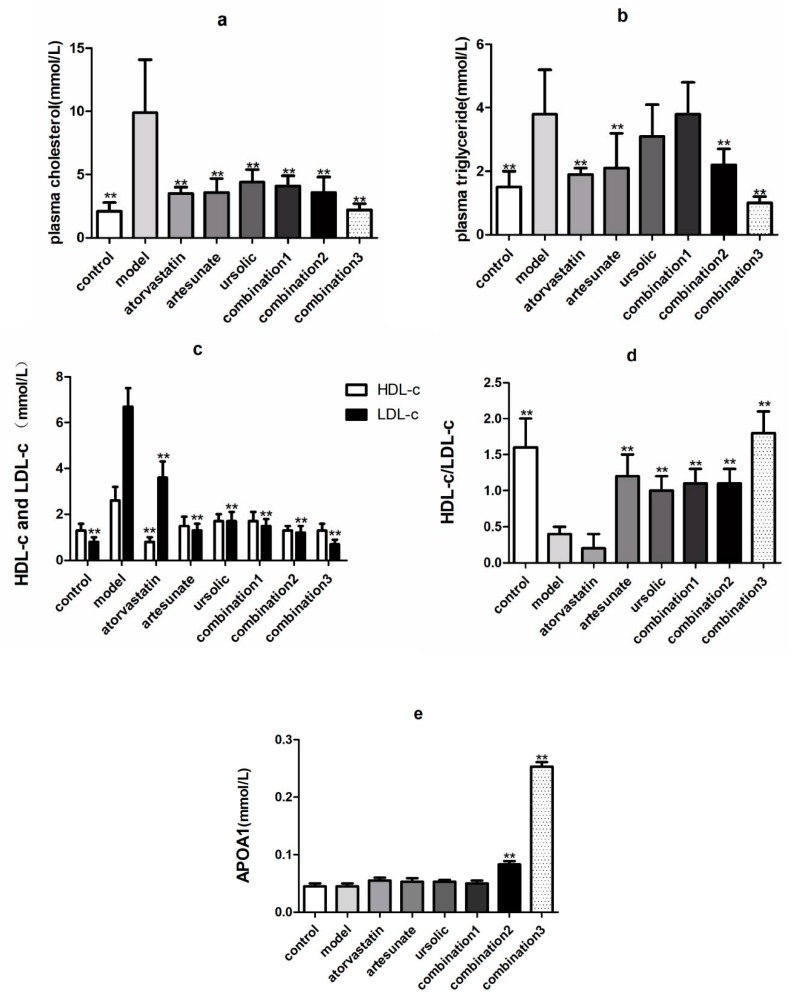
The effect of each administration group on lipid-related metabolites in high-fat rats. The groups were divided as eight groups: control group; model group; positive control (atorvastatin group, 10 mg/kg); artesunate group (20 mg/kg); ursolic acid group (20 mg/kg); combination group 1 (artesunate 10 mg/kg + ursolic acid 10 mg/kg); combination group 2 (artesunate 20 mg/kg + ursolic acid 20 mg/kg); and combination group 3 (artesunate 40 mg/kg + ursolic acid 40 mg/kg). Values are represented as mean ± SD. Significant difference vs. the model group (** *p* < 0.01). (**a**) showed the serum level of cholesterol in each group; (**b**) showed the serum level of triglyceride in each group; (**c**) showed the levels of HDL-C and LDL-C in the plasma in each group; (**d**) showed the ratio of levels of HDL-C to LDL-C in the plasma in each group; (**e**) showed the serum level of APOA1 in each group.

**Figure 2 molecules-25-01392-f002:**
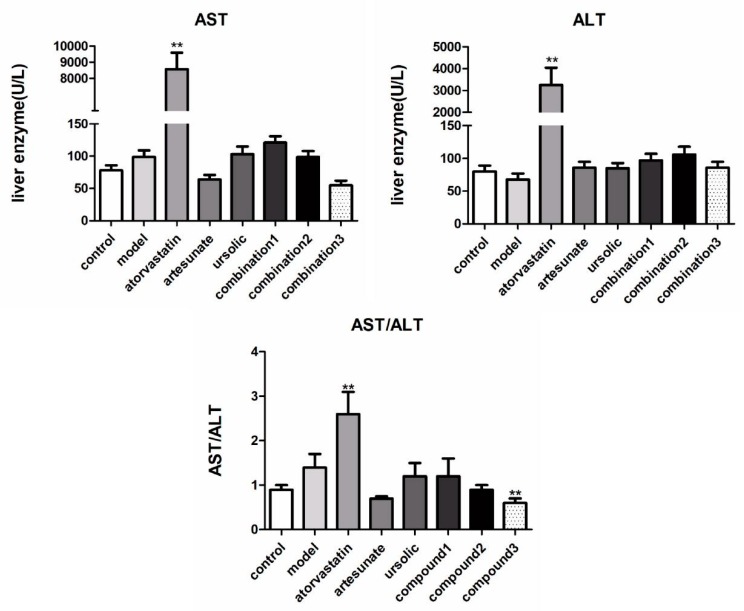
The effect of each administration group on alanine aminotransferase (ALT) and aspartate aminotransferase (AST) in high-fat rats. The groups were divided as eight groups: control group; model group; positive control (atorvastatin group, 10 mg/kg); artesunate group (20 mg/kg); ursolic acid group (20 mg/kg); combination group 1 (artesunate 10 mg/kg + ursolic acid 10 mg/kg); combination group 2 (artesunate 20 mg/kg + ursolic acid 20 mg/kg); combination group 3 (artesunate 40 mg/kg + ursolic acid 40 mg/kg). Values are represented as mean ± SD. Significant difference vs. the model group (** *p* < 0.01).

**Figure 3 molecules-25-01392-f003:**
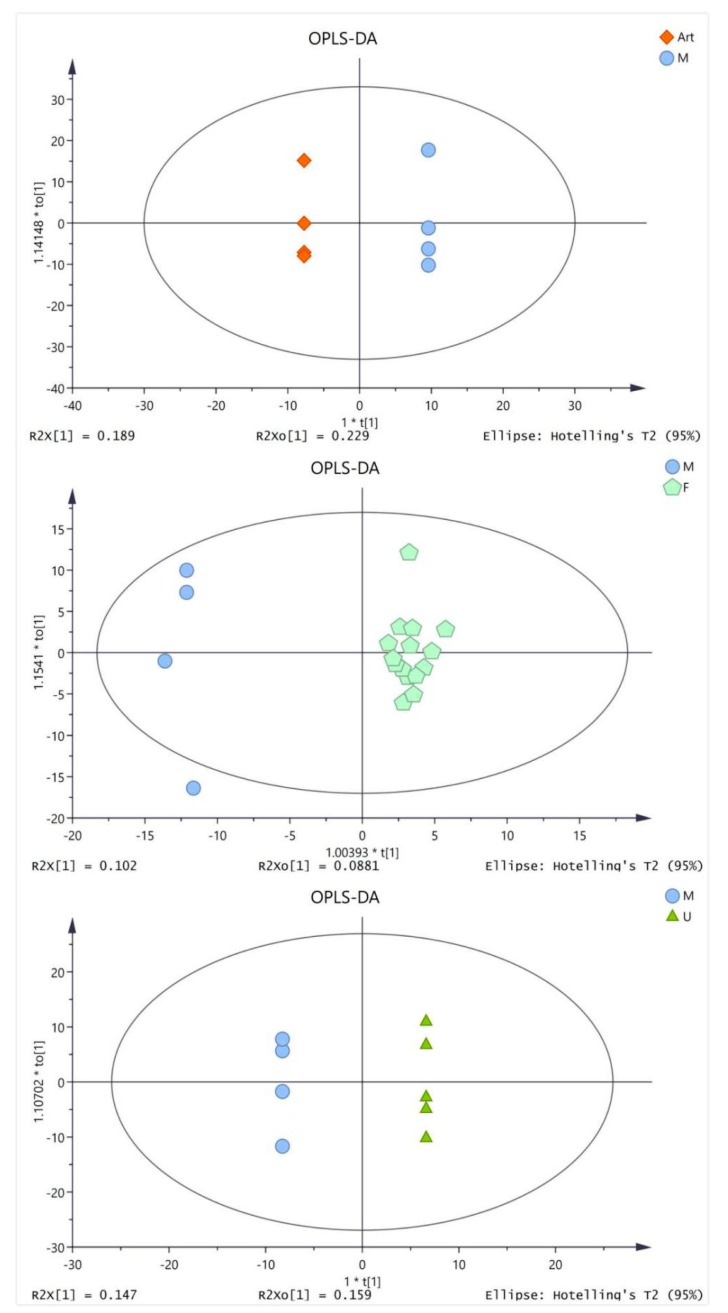
Orthogonal partial least squares discriminant analysis (OPLS-DA) score plots based on the gas chromatography time-of-flight mass spectrometry (GC/TOFMS) data for the model (M) groups vs. artesunate (ART) group (20 mg/kg); ursolic acid (U) group (20 mg/kg) and combination (F) group (artesunate 20 mg/kg + ursolic acid 20 mg/kg).

**Figure 4 molecules-25-01392-f004:**
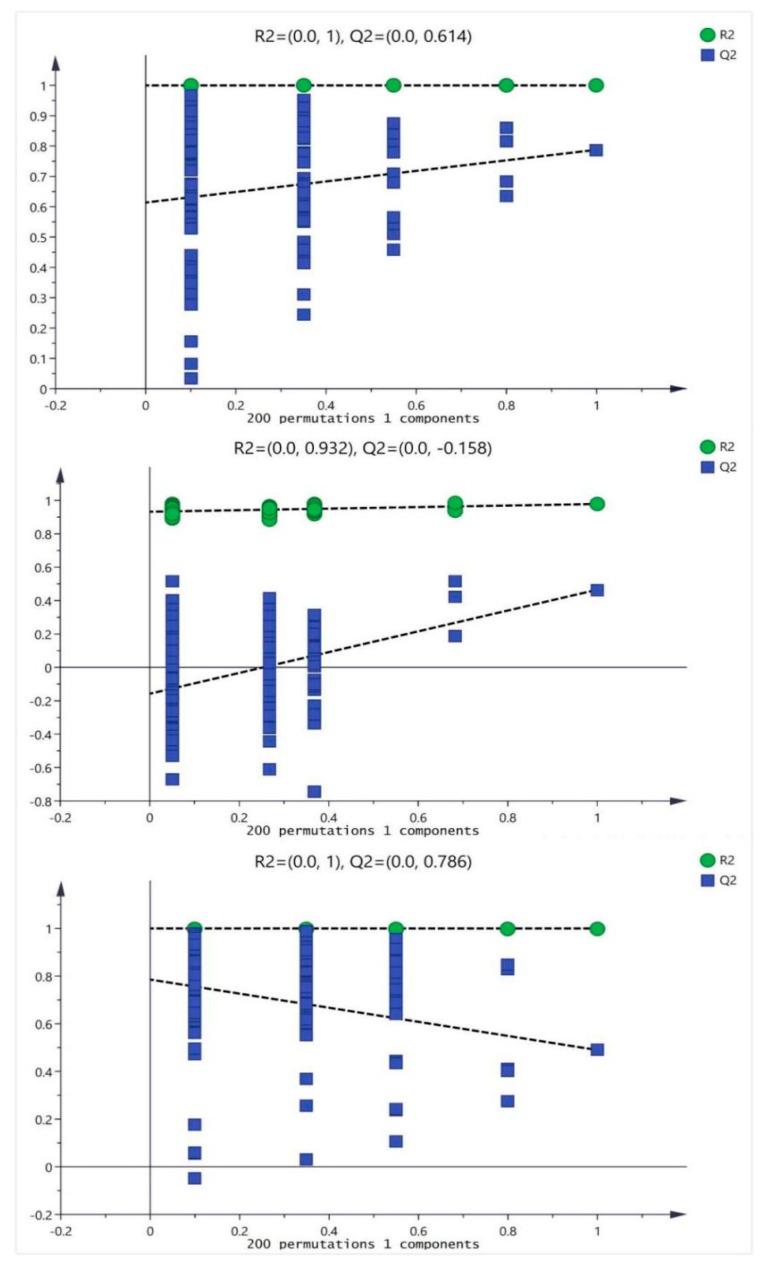
The validation plots of the OPLS-DA models, generated from the seven-cycle interactive verification and 200-response ranking tests with the corresponding predictive principal component. The green square is R2 (cum), denoting the explained variance of the model. The blue diamond is Q2 (cum), standing for the predictive ability of the model. The three pictures from top to bottom represented the artesunate (ART) group (20 mg/kg); ursolic acid(U) group (20 mg/kg) and combination (F) group (artesunate 20 mg/kg + ursolic acid 20 mg/kg), respectively.

**Figure 5 molecules-25-01392-f005:**
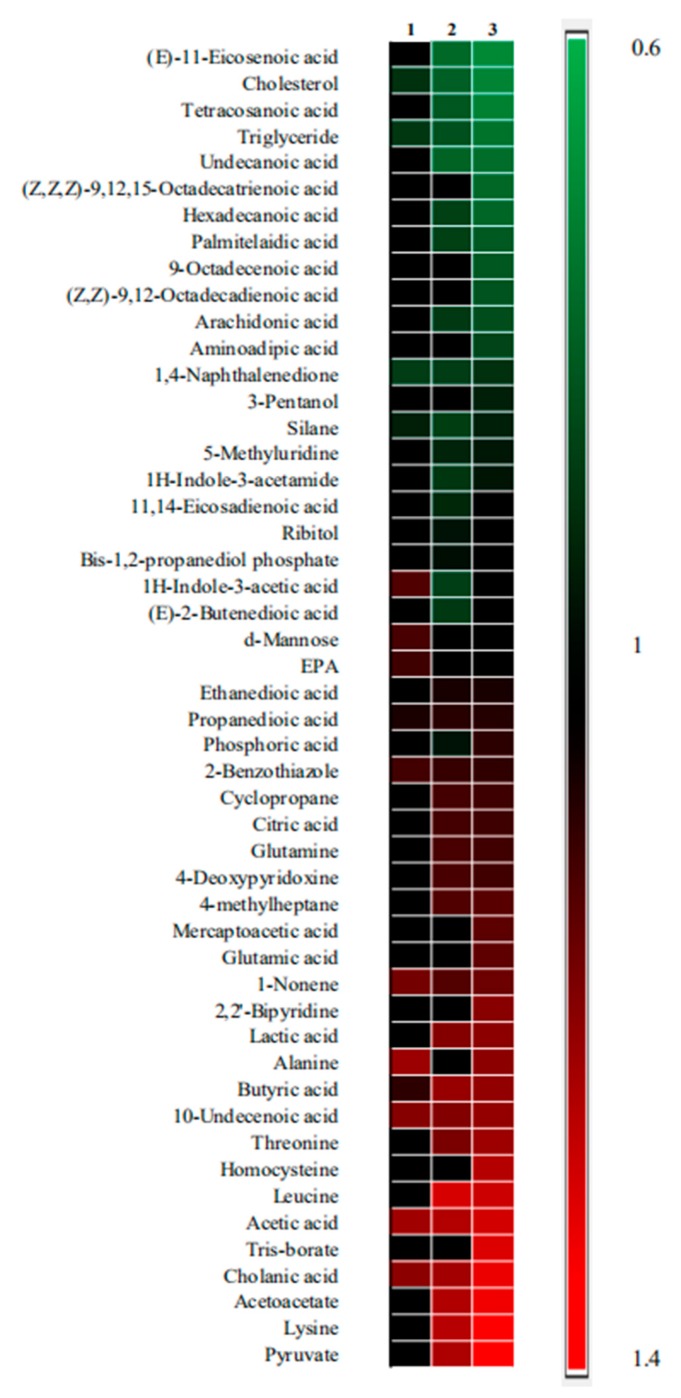
The heatmap of three drug treatment groups compared with the model group. Metabolites with significant differences could be selected according to the criteria of the variable importance in the projection (VIP) > 1.0 and *p* < 0.05 in the t-tests. Moreover, 1 represents the ursolic acid group vs. model group. Furthermore, 2 represents the artesunate group vs. the model group and 3 represents the combination group (ursolic acid and artesunate) vs. the model group. Shades of red represent the fold of metabolite increase compared with the model group; green shades represent the fold of metabolite decrease compared with the model group, respectively. Moreover, black shades represent no significant change in the fold of the metabolite compared with the model group.

**Figure 6 molecules-25-01392-f006:**
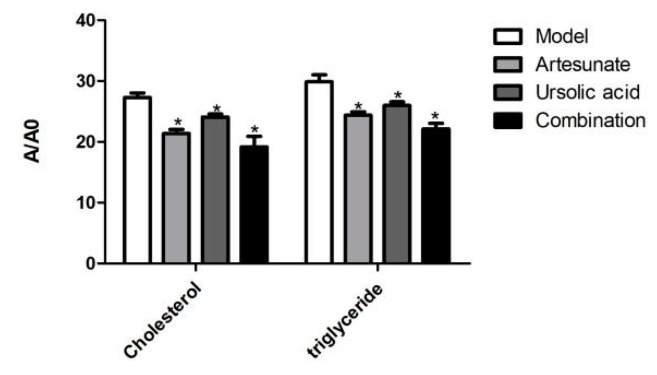
The relative intensity of cholesterol and triglycerides of each administration group in the serum. The groups were divided as four groups: model group; artesunate group (20 mg/kg); ursolic acid group (20 mg/kg); combination group (artesunate 20 mg/kg + ursolic acid 20 mg/kg). A/A0 is expressed by its peak area normalized against that of the internal standard. Values are represented as mean ± SD. Significant difference vs. the model group (* *p* < 0.05).

**Figure 7 molecules-25-01392-f007:**
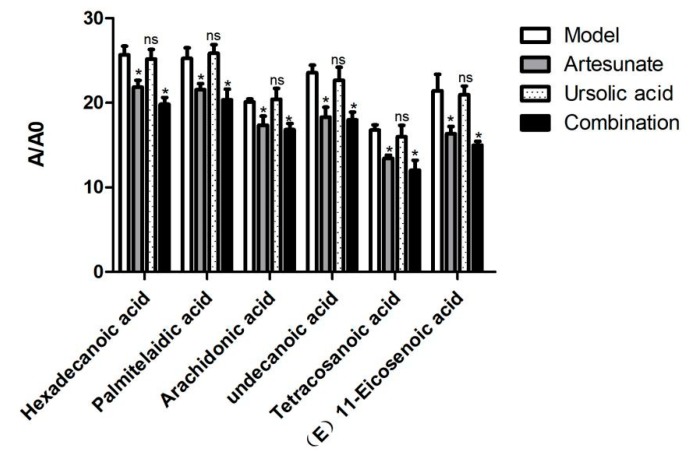
The relative intensity of long chain fatty acids of the model group, artesunate group, and combination group (artesunate + ursolic acid) in the serum. A/A0 is expressed by its peak area normalized against that of the internal standard. Values are represented as mean ± SD. Significant difference vs. the model group (* *p* < 0.05, ns means there was no significant difference between the drug treatment groups and model groups).

**Figure 8 molecules-25-01392-f008:**
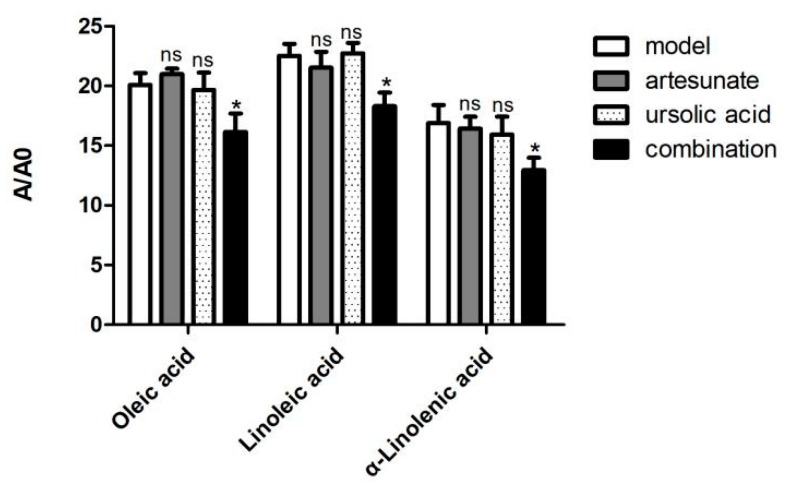
The relative intensity of oleic acids, linoleic acid, and α-linolenic acid of the model group and combination group (artesunate + ursolic acid) in the serum. A/A0 is expressed by its peak area normalized against that of the internal standard. Values are represented as mean ± SD. Significant difference vs. the model group (* *p* < 0.05, ns means there was no significant difference between the drug treatment groups and model groups).

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
