# Peer review of "The Metabolic Changes of Artesunate and Ursolic Acid on Syrian Golden Hamsters Fed with the High-Fat Diet"

_molecules, 2020, doi:10.3390/molecules25061392_

Round 1
Reviewer 1 Report
The authors have pursued further the influence of artesunate and ursolic acid on the detectable metabolites in the plasma samples of Syrian Golden hamster, treated as a model for obesity, by feeding the animals with a high fat diet.
The study is interesting and could provide very important additional information on the undestanding the metabolites increase when high fat diet is applied. However, for the sake of better readability the English language has to be significantly improved.
In addition there a quite a number of correcttions and clarifications which the authors should provide for the better understanding of their work:
- Abstract: the abstract text leads to the assumption that the metabolomic pathways have been investigated. This is unfortunately not the case, as the authors more speculate on the potential metabolomic pathways based on the observed increase or decrease of the particular metabolite concentration. This is quite misleading for the reader, as then the expectations are not fully met.
- Materials and methods: In section 2.1 Chemical and reagents, the text is written in singular form. Here, is shall be clear: how many animals were used, here the authors refer to one hamster - this is unlikely to be the case for a metabolomics study, please, specify the number of animals used in the study.
- Materials and methods: the order of the sections in Materials and Methods is not optimal. The sample preparation for the analysis is described before the mice feeding and sample collection is described. Please, change
- Line 164 refers to Table. S1: Supplementary Information is missing? Please, add the referred to Table.
- Text in lines 186-196 and Fig. 5: The authors should provide an explanation or information about the methodology used to identify the metabolites in the heatmap. The proper place for such an explanation is earlier in the text between the sample measurement and the beginning of the statistical evaluation.
- Information about the exact conposition of the high fat diet is missing and must be provided in the materials and methods section.
The discussion part needs to be improved as the major outcome of this study is lost into the speculation which potential metabolomic mechanism could be affeted without being able to provide scientific proofs for those spectulations. I don't mean to have these speculations removed from the discussion section, the authors have clearly pointed out that they have made estimations, however, I am missing the discussion besed on the real proven outcome from the study.
The conclusion more adequately explains the outcome of the manuscipt as compared with the abstract.
Author Response
- Abstract: the abstract text leads to the assumption that the metabolomic pathways have been investigated. This is unfortunately not the case, as the authors more speculate on the potential metabolomic pathways based on the observed increase or decrease of the particular metabolite concentration. This is quite misleading for the reader, as then the expectations are not fully met.
R: Thank you for this comment. We have changed the abstract in line 12 to 20. The revised abstract is as follows:
Abstract: Artesunate was well known as an antimalarial drug. Our previous research found that it has hypolipidemia effect in rabbit fed with a high-fat diet, especially combined with ursolic acid. In this study, we reconfirmed the lipid-lowering effect of artesunate and ursolic acid in the hamster, and analyzed the metabolic changes using GC/TOF MS. Compared with the model group, a variety of different metabolites of artesunate and ursolic acid alone or in combination were found and confirmed. These differential metabolites include fatty acids, lipids, and amino acids and are involved in lipid metabolism, energy metabolism, and amino acid metabolism. It indicated that two agents of artesunate and ursolic acid could attenuate or normalize the metabolic transformation with different degrees on these metabolic pathways.
- Materials and methods: In section 2.1 Chemical and reagents, the text is written in singular form. Here, is shall be clear: how many animals were used, here the authors refer to one hamster - this is unlikely to be the case for a metabolomics study, please, specify the number of animals used in the study.
R: Thank you for this comment. We have added the number of golden hamsters in line 52.
- Materials and methods: the order of the sections in Materials and Methods is not optimal. The sample preparation for the analysis is described before the mice feeding and sample collection is described. Please, change.
R: Thank you for this comment. We have changed the order of the sections in Materials and Methods.
- Line 164 refers to Table. S1: Supplementary Information is missing? Please, add the referred to Table.
R: Thank you for this comment. We have put the table. S1 in the supporting information.
- Text in lines 186-196 and Fig. 5: The authors should provide an explanation or information about the methodology used to identify the metabolites in the heatmap. The proper place for such an explanation is earlier in the text between the sample measurement and the beginning of the statistical evaluation.
R: Thank you for this comment. We have added the “Metabolite identification was executed by ChromaTOF software combined with National Institute of Standards and Technology (NIS) mass spectral libraries (NIST 14), and LECO/Fiehn Metabolite mass spectral library (Version 1.00) with a similarity of more than 70%. A majority of the metabolites detected were identified using the reference compounds available.”in 2.5 Identification of Metabolites in line 112-118.
- Information about the exact conposition of the high fat diet is missing and must be provided in the materials and methods section.
R: Thank you for this comment. We have added the “The composition of the high-fat diet used in this experiment were listed as follows: lard 10%, cholesterol 2%, propylthiouracil 0.2%, pig bile salt 0.5%, and basic feed 87.3%.” in line 70-71.
Based on your suggestion, we have modified the discussion part to make the major outcome of this study clear.
Reviewer 2 Report
In this manuscript, Pu et al investigated the metabolic changes induced by artesunate and ursolic acid in golden hamsters fed with the high-fat diet. The authors found that artesunate and ursolic acid showed lipid-lowering effects. I have a few questions which I would like the authors to address.
- The authors mentioned that the hypolipidemia effect was observed in rabbits fed with a high-fat diet. In this study, the authors used hamsters. I would appreciate if the authors could explain more about the rationale of testing the same compounds in a different model, and what’s the new findings from current study that would significantly increase our knowledge rather than what’s already been demonstrated in the previous rabbit study.
- For the GC-MS data-acquisition, were there any quality control samples, and blank samples run?
- For the GC-MS data processing, the details on how the authors perform the compound identification are needed. Any software used for the compound identification? What database was used? Was the retention time or retention index examined?
- Was principal component analysis performed on the metabolomics data?
- What’s the rationale of choosing a VIP greater than as the threshold?
- What’s the p-values adjusted for the metabolomics data? What’s the fold change?
- I would appreciate if the authors indicate the number of replicates used in each experiment in the figure legends.
Author Response
Dear reviewer,
Please see the attachment.
Thanks,

Reviewer 3 Report
This is an interesting paper on the effect of artesumate and ursolic acid on blood lipids.
There is one major and many minor concerns.
Abstract
Line 20 and 21: These lines overstate the results. More on this below.
Introduction
Line 29: The following sentence is unclear in meaning, “Artesunate is a semisynthetic derivative of artemisinin, have stronger anti-malaria effect[8-10].” Please clarify.
Line 45: Change “are” to “is”
Line 48: Change “method explored” to “method we explored”.
Line 49: Change “understand” to “understanding”.
Materials and Methods
Line 54: Remove “And”.
Line 67: I suggest changing “detection” to “analysis”.
Lines 70 and 71: Change “respectively. Using” to “respectively, using”.
Line 78: Change “hamsters, received” to “hamsters and received”.
Line 79: Remove “respectively”.
Line 80: Remove “which”.
Line 83: Include words for abbreviations ALT and AST.
Line 83: Change “measured in the hereafter” to “were subsequently measured”.
Line 84: Change “ determination” to “determinations”
Line 85: Change “A value” to “A p-value”.
Line 94: Change “make the residue derivatize” to “derivatize the residue”.
Line 99: Remove “which”.
Line 103: Change “incipient” to “initial”.
Line 104: Change “equably raised” to “raised linearly”.
Line 106: Change “severally” to “respectively”.
Line 107: Remove “of”.
Line 111: Change “softare” to “software”.
Line 112: The MathWorks address is missing a city.
Line 122: Change “whose” to “with”.
Other grammatical changes may be needed in the remainder of the text.
Line 134: While it appears visually in Figure 1c that the ratio of HDL to LDL is altered by the treatment, the graph itself does not plot the ratio and, of course, no statistics were done on the ratio. What is apparent from Figure 1c is that none of the treatments had a statistically significant effect on HDL. So, the authors should either state the no-effect result or change or add a graph of the ratio.
Line 135: This is a major critique of the entire paper and it is shown for the first time here. The combination treatment is not the best hypolipidemic treatment. The comparable combination treatment with the same dose of artesunate and ursolic as the individual dose (combination 2) has visually about the same effect as the individual treatments for total cholesterol and triglycerides and slightly lower LDL and HDL and quite higher APOA1. This is quite unimpressive as the “best hypolipidemic treatment.” But even more to the point, this particular hypothesis, that the combined treatments are better than the mono-treatments was never tested statistically. And while the double dose combination 3 may have superior effects, there is roughly 4 times as much medicine given to those animals, so the results are not surprising and comparing them with low dose mono-treatments is unimpressive.
There are several interesting and impressive results from this study that are worth noting. Both agents have rather astonishingly hypolipidemic effects, closely matching one of the most effective statins on the market. Their double dose combination also greatly increases APOA1 over mono-treatments. They also have lower liver toxicity than atorvastatin. The combination also lowers several key fatty acids.
Figure 7 needs the Ursolic treatment group.
Lines 221 and 222: Again, the authors are overstating their results. Be more specific and remove this text.
Figure 8 needs the mono-treatment groups.
Lines 294 and 295: Again, the authors are overstating their results. Be more specific and remove this text.
Author Response
For Reviewer 3:
1.Abstract:Line 20 and 21: These lines overstate the results.
R: Thank you for this comment. We have modified the abstract in line 17 to 20.
2.Introduction:Line 29: The following sentence is unclear in meaning, “Artesunate is a semisynthetic derivative of artemisinin, have stronger anti-malaria effect[8-10].” Please clarify.
R: Thank you for this comment. We have changed the “stronger” to “efficient” in line 28. Here we just express that artesunate has anti-malaria effects.
3.Line 134: While it appears visually in Figure 1c that the ratio of HDL to LDL is altered by the treatment, the graph itself does not plot the ratio and, of course, no statistics were done on the ratio. What is apparent from Figure 1c is that none of the treatments had a statistically significant effect on HDL. So, the authors should either state the no-effect result or change or add a graph of the ratio.
R: Thank you for this comment. We have added the figure about the ratio of HDL-C/LDL-C as the figure 1d.
4.Line 135: This is a major critique of the entire paper and it is shown for the first time here. The combination treatment is not the best hypolipidemic treatment. The comparable combination treatment with the same dose of artesunate and ursolic as the individual dose (combination 2) has visually about the same effect as the individual treatments for total cholesterol and triglycerides and slightly lower LDL and HDL and quite higher APOA1. This is quite unimpressive as the “best hypolipidemic treatment.” But even more to the point, this particular hypothesis, that the combined treatments are better than the mono-treatments was never tested statistically. And while the double dose combination 3 may have superior effects, there is roughly 4 times as much medicine given to those animals, so the results are not surprising and comparing them with low dose mono-treatments is unimpressive.
R: Thank you for this comment. We have removed “Combination of artesunate and ursolic acid has the best hypolipidemic effect, and is the only one that can increase the apoA-I content of serum”, instead of “The high-dose combination group 2 and 3 were the only two that can increase the apoA-I content of serum.” in line 150-151.
- Figure 7 needs the Ursolic treatment group.
R: Thank you for this comment. As we mentioned in the heat map(Fig.5), no long chain fatty acids were selected in ursolic treatment group based on the criterion of VIP>1 and p-value <0.05.
- Figure 8 needs the mono-treatment groups.
R: Thank you for this comment. As we mentioned in the heat map(Fig.5), the oleic acid, α-linolenic acid and linoleic acid in ursolic treatment groups and artesunate treatment groups were not selected based on the criterion of VIP>1 and p-value <0.05.
7.Lines 221 and 222: Again, the authors are overstating their results. Be more specific and remove this text.
R: Thank you for this comment. We have removed “This is probably the reason why the treatment effect of combination was more significant than artesunate or ursolic acid” in line 233-234.
8.Lines 294 and 295: Again, the authors are overstating their results. Be more specific and remove this text.
R: Thank you for this comment. We have removed “and the combination of two drugs are better than either agent alone.”.
Round 2
Reviewer 1 Report
The authors have presented a signifficantly improved version of the manuscript and despite of mainly readability issues, including further need for English language corrections, the manuscript deserves to be accepted for publication.
Author Response
Thank you for your comments. We have tried our best to improve the readability with the help of other colleagues.
Reviewer 2 Report
The authors addressed most of my comments. I still have two minor comments which I would like the authors to address.
- Adjusted p-values need to be provided for the metabolomics data to adjust the multiple comparison.
- The authors mentioned that “To minimize systematic analytical deviations, each analytical batch may contain one spiked sample as quality control and one blank sample run together every 10 injections.” Since the quality control samples were run, relative standard deviations need to be calculated for the metabolites in the spiked quality control samples to evaluate the analytical deviations. If the relative standard deviations of the metabolites in quality controls are relatively large, the threshold for fold change >1.06 or < 0.94 might not be suitable as the fold change may not be caused by the treatment.
Author Response
1.Adjusted p-values need to be provided for the metabolomics data to adjust the multiple comparison.
R: Thank you for this comment. As we mentioned in line 135, the p-values were acquired from t-tests. And the t-test is used for comparison between pairs. Therefore, we just compare the different drug groups with the model group adopting the t-test, respectively. But we didn't do multiple comparisons between drug treatment groups using the t-test. We also avoided comparing the curative effects of different drugs in our discussion and conclusion.
2.The authors mentioned that“To minimize systematic analytical deviations, each analytical batch may contain one spiked sample as quality control and one blank sample run together every 10 injections.” Since the quality control samples were run, relative standard deviations need to be calculated for the metabolites in the spiked quality control samples to evaluate the analytical deviations. If the relative standard deviations of the metabolites in quality controls are relatively large, the threshold for fold change >1.06 or < 0.94 might not be suitable as the fold change may not be caused by the treatment.
R: Thank you for this comment. According to your advice, we changed the threshold for fold change to FC>1.20 or FC<0.80. The reason for choosing this threshold is that a 20% gap is usually considered significant. In addition, in recent years, the selected threshold of fold change in some metabolomics articles was also FC>1.20 or FC<0.80. And these metabolomics articles are listed at the end of the paragraph.
The lipids, the long-chain fatty acids and the amino acids we mentioned in the discussion part were not affected by the change of the new threshold. Based on the criterion of FC>1.20 or FC<0.80, there were 28 differential metabolites in the combination group, 19 metabolites in the artesunate group and 8 in the ursolic acid group compared with the model group.
- Zhao L, Wu H, Qiu M, et al. Metabolic Signatures of Kidney Yang Deficiency Syndrome and Protective Effects of Two Herbal Extracts in Rats Using GC/TOF MS[J]. Evid Based Complement Alternat Med, 2013: 1-10.
- Liu Y, Chen T, Qiu Y, et al. An ultrasonication-assisted extraction and derivatization protocol for GC/TOFMS-based metabolite profiling[J]. Anal Bioanal Chem, 2011, 400(5):1405-17.
Reviewer 3 Report
This manuscript has been significantly improved and the major concern of my original review has been addressed. The text still needs considerable assistance with its English and with these few remaining minor concerns.
Line 154: Remove “HDL-c”. From Figure 1 we can see that HDL-c in the model group is not significantly higher than the control group.
Figure 7 still needs the Ursolic treatment group.
Figure 8 still needs the mono-treatment groups.
Line 291: Change “catalyzed” to “metabolized”
Lines 312 and 313: Change “Glutathione (GSH) was a momentous antioxidant in cells, whose main function was to protect biofilms, nucleotides and various proteins from free radical damage.” to “Glutathione (GSH) is an important antioxidant in cells, whose main function is to protect biofilms, nucleotides and various proteins from free radical damage.” When stating presently known facts such as this, they can be stated in the present tense.
Author Response
Thank you for your advice, we have revised the mistake and add the missed group in the fig7 and fig 8.